# Field Evaluation of Rice Lines Derived from Suakoko 8 X Bao Thai for Iron Tolerance in the South Saharan African Farming System

**DOI:** 10.3390/plants13121610

**Published:** 2024-06-11

**Authors:** Mouritala Sikirou, Afeez Shittu, Yonnelle Dea Moukoumbi, Aboudou Hack Arouna, Chédrac Zokpon, Roland Bocco, Adetoro Najimu, Venuprasad Ramaiah

**Affiliations:** 1International Institute of Tropical Agriculture, 4163 Av. Haut Congo, C/Gombe, Kinshasa, Democratic Republic of the Congo; 2Laboratoire des Sciences Végétales, Horticoles et Forestières, School of Horticulture and Green Landscaping, National University of Agriculture, Kétou P.O. Box 043, Benin; 3International Rice Research Institute (IRRI), Pili Drive, P.O. Box 7777, Los Baños 4031, Laguna, Philippines; 4Institut de Recherches Agronomiques et Forestières (IRAF), Libreville P.O. Box 16169, Gabon; 5Department of Agriculture and Natural Resources (ANR), UC Davis, Cooperative Extension, 2156, Sierra Way, Ste C, San Luis Obispo, CA 93401, USA; rolandbocco@gmail.com

**Keywords:** lowland rice, lines, correlations, leaf bronzing score, yield, iron-toxicity

## Abstract

Rice is a major grain crop in numerous countries. In lowland areas, high iron levels in the soil severely hinder its cultivation. The current study explored high-yielding and Fe-toxicity-tolerant irrigated lowland rice (340 lines) among a population derived from a cross between Suakoko 8 and Bao Thai in Edozighi and Ibadan, Nigeria. In contrast to Ibadan, the soils in Edozighi contain a significant amount of iron. For the stated purpose, we carried out a two-year experiment using an alpha lattice design. The data showed significant differences between genotypes for the days to heading, plant height, number of tillers per plant, number of panicles per plant, panicle length, and grain yield. The results revealed that multiple characteristics had both direct and indirect effects on cultivated rice yields. There was a direct and positive influence on the number of days in the 50% heading period (0.31), a direct and negative effect on plant height (−0.94), a direct and positive effect on tiller and panicle numbers, and a direct but negative effect on panicle length (−0.56). The leaf bronzing score was adversely correlated with yield, panicle length, and plant height, while it was positively correlated with the number of panicles, tillers, and days to heading. The findings showed significant changes in yield and yield characteristics between genotypes. Grain yields ranged from 283 to 11,700 kg/ha in the absence of iron in the soil, contrary to 0 to 8230 kg/ha in soil with iron toxicity, with losses estimated between 6 and 94%, demonstrating the resulting disaster. In contrast to the elite parents and varieties used in this study, the ten top genotypes exhibited smaller losses in yield. The authors strongly recommend using these lines for further studies as donors or releasing them in farmer fields in Africa.

## 1. Introduction

In Western Africa, rice is a crucial food security staple [1,2]. Due to population expansion, urbanization, and rising demand, rice consumption in this region has rapidly expanded [3]. The sub-region’s average per-capita consumption rose from 32 kg in 1990 to 34 kg in 2000 before peaking at 49 kg in 2012 [4]. In West Africa, where local production only meets 60% of the region’s needs, there is regrettably still a constant rice shortage. Rice self-sufficiency rates are less than 40% in nations like Senegal, Ghana, Benin, and Côte d‘Ivoire [5]. According to estimations for 2011 and 2012, regional imports totaled between 7 and 8 million tons, costing between USD 3.5 and USD 4.0 billion [6]. More than half of West Africa’s rice imports come from Nigeria, Senegal, Côte d‘Ivoire, and Benin [7]. According to Fitzgerald et al. [8], rice is the most significant crop among the essential carbohydrate sources that feed the world’s population, providing over 21% of the caloric demands of the overall population. Researchers cite abiotic and biotic stresses as the main factors that reduce yield in rice [9].

Most organisms require iron for normal growth. As a chelating molecule, iron (Fe) is essential for plant metabolic processes like mitochondrial respiration, photosynthesis, electron transport, and other redox reactions [10,11]. Iron toxicity, on the other hand, can result from excessive Fe buildup in cells, causing nutritional problems, physiological and agronomic decline, and even plant mortality, Li et al., 2024 [12,13]. One of the most significant abiotic pressures on rice productivity in lowland areas in many parts of the world is Fe toxicity, and Mahender et al. [14] found that this stress affects about 18% of soils worldwide. 

Depending on the rice genotype, the quantity of Fe contamination, and the soil fertility level, decreases in rice production under Fe stress ranged from 12 to 100% [15,16]. Within the oxygen-poor conditions of tropical lowland rice fields, ferric oxide (Fe^3+^) changes into the reduced ferrous form of Fe^2+^. This makes more Fe toxicity available [12]. According to several studies [17,18], the extra Fe^2+^ is transferred from the root to the shoot and causes leaf bronzing, cellular oxidative harm, nutritional deficit, and reduced rice development. Rice exhibits several defensive strategies against Fe toxicity in diverse ways at different developmental stages [17]. According to Rasheed et al. [12], several genes have been linked to numerous physiological and agronomic reactions to Fe toxicity, including Fe^2+^ absorption, translocation, subcellular translocation, and Fe^2+^ regulation. A fantastic answer to the Fe toxicity issue is the breeding of Fe^2+^-tolerant rice varieties. In order to release rice varieties that are tolerant/resistant to Fe^2+^, a marker-assisted breeding method is a promising approach. Rice intra-specific and inter-specific mapping populations have been used in a number of studies to try and discover quantitative trait loci (QTLs) or genes linked with Fe^2+^ toxicity [19,20,21]. The identification of QTLs and genes linked to Fe^2+^ toxicity-related depictions in rice has also been performed using genome-wide association studies [21,22,23].

To test plant material in multiple environments, many plant breeders conducted multi-environment trials to develop varieties adapted to various conditions [24]. Multi-environment trial data have helped discover high-yielding maize, sorghum, irrigated, and upland rice varieties [25,26,27,28,29,30]. Classical statistical models analyze genotype–environment interactions and provide important information. These models include site regression, genotype regression, shifted multiplicative, totally multiplicative, and genotype-by-environment biplot. Additive Main Effect, and Multiplicative Interaction (AMMI), and Genotype × Genotype × Environment (GGE) are popular with plant breeders. AMMI used analysis of variance and principal component analysis [31], while GGE biplot used genotype and environment scores. The AMMI model only captures genotype by environment using singular value decomposition on twice-centered, two-way data. However, in environment-centered two-way data [32], GGE uses singular value decomposition to capture genotype and genotype by environment. Genotype-by-environment models have been used in upland rice breeding for drought stress resistance [33,34], adaptation and yield stability [35], optimum planting date [36], and root morphology [37]. 

The current study explored high-yielding and Fe-toxicity-tolerant irrigated lowland rice among a population derived from crossing Suakoko 8 and Bao Thai under field conditions.

## 2. Results

### 2.1. Weather Patterns at the Screening Sites

According to the first assessment, there was no difference in rainfall between Ibadan and Edozighi in July 2012 (Figure 1). August was rainier in Edozighi, while the rest of the months were rainier in Ibadan. In 2013, Ibadan was the rainiest during the whole evaluation period. The average temperature was the same (*p* < 0.05) between Ibadan and Edozighi from July 2012 to September 2012 and 2013, but the heat recorded in Edozighi was higher until the end of the two experiments (Figure 1).

### 2.2. Probability Study

Shapiro–Wilk normality tests on genotype growth characteristics yielded *p*-values under 0.001 (Table 1). Thus, the quantitative variable’s normalcy assumptions are rejected with 99.99% confidence. In this scenario, the Kruskal–Wallis test serves as an alternative to the analysis of variance. There were significant differences between genotypes in the number of days until 50% heading (DH), plant height (PH), number of stems per plant (NT), number of panicles per plant (NP), panicle length per plant (PL), and yields by genotype in kg/ha (GY). The Kruskal–Wallis tests on the median values of the studied genotype growth characteristics are shown in the Table 1.

### 2.3. Mean Variabilities

The analysis of the variables’ distributions displays different patterns observed in all prior quantitative parameters studied. Table 2 presents the summary and findings of numerous identified trends. The average period from sowing to heading at the Ibadan location was 85.7 ± 8.79 days after sowing (DAS), with a range of 62.0 to 110 DAS. The duration in Edozighi ranged from 69 to 116 DAS, with an average of 91.8 ± 9.85 days. Days from sowing to heading (DH) fluctuated between 62 and 116, with an average of 88.7 ± 9.82 days. The median analysis showed that more than half of the plants had reached semi-flowering 90 days following plantation.

Plant heights in Ibadan ranged from 102 to 201 cm with an average of 130 ± 17.5 cm, while in Edozighi they ranged from 58.3 to 148 cm with an average of 108 ± 12.6 cm. The average height at both sites is 134 ± 30.5 cm, with extreme values ranging from 58.33 to 201 cm, including a median height of 127 cm. Plants were taller in Ibadan than in Edozighi, characterized by an abundance of iron in the soil. High Fe^2+^ levels in the soil in Edozighi resulted in a decrease of around 32.5% in plant height compared to the average recorded in Ibadan (Table 2).

According to the results, Edozighi had more tillers on plants than Ibadan. In Ibadan, tillers ranged from 5.00 to 23.00, with an average of 11 ± 2.97 tillers per plant. However, in Edozighi, the genotypes had 10.00 to 36.30 tillers per plant, with an average of 19.50 ± 4.82 tillers. The population analyzed had tillers per plant ranging from 5.00 to 36.30, with an average of 15.20 ± 5.85 and a median of 14.3. Edozighi had more panicles per plant than Ibadan, whose soil had low iron levels. In Ibadan, the number ranged from 3.33 to 26.7, with an average of 9.47 ± 2.80 panicles per plant. The study found an average of 16.5 ± 4.69 panicles per plant in Edozighi, with a range of 6.67 to 34.3. More than half of the accessions evaluated had more than 12 panicles per plant, with a median of 12.0. There was a 12.5% decrease in panicle length in plants evaluated in a growing condition with a high Fe^2+^ content, as compared to plants sown in Ibadan (Table 2). In Ibadan, panicle lengths ranged from 20.0 to 41.0 cm, with an average of 28.8 ± 3.17 cm. In Edozighi, panicle length varied from 18.3 to 32.0 cm per plant, with an average of 25.2 ± 2.99 cm. Panicles in Ibadan are longer than in Edozighi, measuring 18.3 to 41.0 cm, with the majority exceeding 27 cm.

Plant output is higher in Ibadan than in Edozighi, where plants are more susceptible to Fe^2+^. Yields in Ibadan varied between 283 and 11,700 kg/ha, with an average of 6120 ± 2520 kg/ha. In Edozighi, yields ranged from 0 to 8230 kg/ha, with an average of 1390 ± 665 kg. Moreover, half of the plants evaluated yielded more than 2240 kg/ha, resulting in an average yield of 3750 ± 3000 kg/ha. The excessive Fe^2+^ content in Edozighi reduced the yield by 77.28% reduction in yield when compared to the average yield in Ibadan.

### 2.4. Correlation Analysis

The correlogram illustrates that there are a few substantial relationships at 5% between flower growth traits (Figure 1). The correlation study revealed a high positive association between stem number and panicle number (0.97). Stem and panicle counts, on the other hand, have a negative correlation with grain yield. However, the number of days at 50% heading (DH) has a negative correlation with plant yields, panicle lengths, and plant height. However, the number of days observed at the 50% heading stage has a positive and weak relationship with the number of tillers and panicles, respectively. A highly significant negative correlation was noticed between the Fe toxicity score and plant height (−0.84), panicle length (−0.51), and plant yield (−0.78). However, a significant positive relationship was recorded between Fe toxicity and the number of tillers and panicles. Similarly, we observed a positive association between iron toxicity and the time required for heading.

### 2.5. Path Analysis

The path analysis of the examined variables revealed both direct and indirect effects on cultivated rice yields (Figure 2). According to the analysis, the transition to a high-iron toxicity environment has a direct and positive effect on the number of days in the 50% heading period (0.31), a direct and negative effect on plant height (−0.94), a direct and positive effect on the numbers of tillers and panicles, and a direct but negative effect on panicle length (−0.56).

Furthermore, the transition to a high-iron environment helps to increase the number of tillers (0.73%). This increase in the number of tillers supports an increase in the number of panicles, with a total variance of 75%. However, the panicles obtained had a 56% reduction in length. On the other hand, increasing plant height stimulates an increase in production; the delay during the flowering phase and the small size of the plants had a detrimental influence on plant yield. At that point, the move to an environment with Fe toxicity correlates to a sharp reduction (60.16%) in grain yields.

### 2.6. Classification of Genotypes and Principal Component Analysis

The studied genotypes were divided into four homogeneous groups using hierarchical classification and principal component analysis (Figure 3). The first two groups represented those exposed to environmental iron toxicity, whereas the latter two represented those cultivated in the control environment. Figure 3 summarizes the results of this study. The first category accounted for 20.29% of the tested population. It comprised 98.18% of genotypes grown under conditions characterized by iron toxicity. This group of genotypes had an average height of 111 ± 12.8 cm, 24.1 ± 3.46 tillers, and 21 ± 3.43 panicles. On average, this category’s plants attained 50% heading in 93.7 ± 9.02 days. Their average yield per hectare was 1550 ± 832 kg. This category included those who produce more grains. The second group included 30.88% of the study’s individuals. About 96.90% of genotypes grew in an environment with Fe toxicity. This group of genotypes had an average of 13.4 ± 2.51 panicles and 16.2 ± 2.66 tillers. These genotypes had a median size of 107 cm and produced an average of 1310 ± 643 kg/ha. This group included genotypes with low yields at the end of the trial.

The third category comprised 15.44% of the study’s genotypes. About 99.09% of the accessions planted at the Ibadan control site were what set it apart. These individuals had roughly 9.86 ± 3.09 panicles and 11.4 ± 3.01 tillers, with a median plant height of 146 ± 17.0 cm and an average yield of 5570 ± 2110 kg/ha. These plants had an average panicle length of 29.3 ± 3.03 cm. This category included cultivars with low grain yields from the start. The fourth category represented 33.38% of the genotypes in the study. This category encompassed all genotypes grown at Ibadan’s control site. This group had around 9.17 ± 2.40 panicles and 10.6 ± 2.67 tillers, with median and average heights of 168 and 168 ± 12.2 cm, respectively, with an average grain output of 6500 ± 2550 kg/ha. This was the group of accessions that had a high basic yield.

### 2.7. Determination of the Top 10 Performing Genotypes

An in-depth evaluation of the tested population revealed yield losses of up to 94%. The ten best genotypes, their parents, and the controls utilized, together with their yields in kg per ha, are presented in Table 3. Our findings revealed that IR 88638-230-1-1-1-1-1-1 (2855 kg/ha) and IR 88638-39-1-1-1-1-1 (2803 kg/ha) under stress outperformed 4793 and 5223 kg/ha under control, with a yield loss of 40 and 46%, respectively. Under stress, some of these lines, such as IR 88638-34-1-1-1-1-1-1, IR 88638-98-1-1-1-1-1, IR 88638-240-1-1-1-1-1, IR 88638-13-1-1-1-1-1-1, IR 88638-230-1-1-1-1-1, IR 88638-308-1-1-1-1-1-1, IR 88638-39-1-1-1-1-1-1, and IR 88638-208-1-1-1-1-1, had a higher yield than the donor parent of the tolerant gene (Suakoko 8) with 987 kg/ha. Similarly, based on the observations, the three lines IR 88638-325-1-1-1-1-1-1, IR 88638-102-1-1-1-1-1-1, and IR 88638-34-1-1-1-1-1-1 showed a very minor loss in yield at the lack of and with an excess of iron in the soil, with LBS of 6, 7, and 7. Under stress, the chosen lines produced better yields than their two parents and the local elite varieties used as controls. A Appendix A contains a complete list of population yield performances in Edozighi and Ibadan.

## 3. Materials and Methods

### 3.1. Experimental Site and Genetic Materials

The field assessment was carried out in the Nigerian cities of Edozighi and Ibadan in 2012 and 2013, approximately encompassing five months from July to December, which corresponds to the rainy season. Figure 4 shows the weather conditions in the two sites. The town of Edozighi is located in the state of Niger in northern Nigeria. According to Nwilene et al. [38], Edozighi is in the tropical-warm/sub-humid agro-ecological zone, with acidic (pH 4.2–5.2) clay loam soil [39]. In contrast, Ibadan is located in Oyo State, and the trial was conducted in the wetland of the International Institute of Tropical Agriculture (IITA), situated at 7°29′ N and 3°54′ E, with alfisol type and acidic (pH < 6). The Edozighi site is well known for the extreme Fe^2+^ toxicity that afflicts its rice farming, whereas the Ibadan site is distinguished by the lack of symptoms associated with Fe^2+^ in the organs of the plant [40]. The soil properties were the same as reported by Sikirou et al. [40].

For the first test, 340 genotypes of rice were used. These included the F_5_ and F_6_ families of 335 lines that were developed by crossing the elite irrigated resistant rice variety Suakoko 8 and the susceptible one Bao Thai [40,41]. The other five were the controls that comprised the two parents of the families (Suakoko 8 and Bao Thai), BW348-1 and WITA4 (both resistant), which mostly grow in Nigeria, and IR64 (susceptible), which are widely cultivated in several rice crops in Africa and Asia.

### 3.2. Field Management

Deep plowing had been the first step in preparing the land, then harrowing and soil leveling followed. Grasses were uprooted during soil preparation operations but remained on the surface and were manually removed. The fertilizers used were NPK (15, 15, and 15) and urea applied at a rate of 200 kg/ha and 50 kg/ha, respectively. Urea was applied on the day of the transplant, and NPK was provided in two split applications at 50% of the normal rate. The first application was at 42 DAS, and the second was at 63 DAS. These dates marked the peak of tillering and heading for most genotypes.

Prior to transplanting at 14 days after sowing (DAS), at each of the two experimental sites, a seedbed was carefully constructed to serve as a nursery. To prevent mixing accessions, a labeled line represents each genotype. Water was administered twice a day to ensure the proper germination of the seeds. The field transplanting process involved transferring the seedlings, one plant per plot, to their respective elementary plots. The experimental set-up used was an alpha lattice in two replications (each of the two localities) per year. The trial contained a total of 364 elementary plots, divided into 13 incomplete blocks of 28 plots. The parcel area consisted of a single 3 m row where seedlings were transplanted, with a spacing of 0.2 m × 0.2 m as the distance between two consecutive plants and rows. With only 340 genotypes, out of the 24 unoccupied plots, 24 were filled by controls. At the end of the year 1 evaluation, plants that had not succumbed to the Fe^2+^ toxicity disease had been harvested. This previous behavior, added to the quality of the grains and yield, is one of the factors that motivates rice farmers to adopt improved rice commodities (personal communication). The early lines of the family were harvested at 120 DAS, and water maintenance in the plot was stopped at 90 DAS to avoid grain loss due to rotting with excess moisture during irrigation. Harvesting and ginning had been performed manually. Figure 5a,b depict plant appearances after the heading stage in Idaban and Edozighi, respectively.

### 3.3. Data Collection

Agromorphological data were collected at the appropriate growth stage of rice, following field collection techniques [42] and the Standard Rating System (S.E.S.) for rice (IRRI, 2002). Six quantitative traits were evaluated, such as days to heading (days up to 50% heading), plant height, number of tillers, number of panicles, panicle length, and grain yield. The Leaf Bronzing Score (LBS) was collected based on leaf symptoms and general appearance on a scale of 0 to 9 (0 = normal or near-normal plant; 9 = nearly dead or dead plant) using a scale developed at the International Rice Research Institute (IRRI) at 35, 56, and 77 DAS [43].

### 3.4. Statistical Analysis

The open-source statistical analysis software R 4.3.2 was used to carry out the data analysis. We used the ‘stats’ package to do descriptive statistics, the Kruskal–Wallis median analysis test, and a correlation study of traits like plant height (PH), number of tillers (NT), number of panicles (NP), panicle length (PL), and yield in kg per hectare (GY). The ‘ggcorrplot’ package was required for correlogram representation. The ‘FactomineR’ and ‘factoextra’ packages were called for the principal component analysis (PCA) of our study’s quantitative variables, as well as the creation of the hierarchical classification model. Path analysis was used to evaluate direct and indirect interactions among a variety of variables of interest. The packages ‘lavaan’, ‘semPlot’, ‘OpenMx’, and ‘GGally’ were required for this analysis. At the end, the formula of Reyniers et al. [44] was used to separate genotypes that performed best in terms of grain yield under stress.

## 4. Discussion

As a result of pollution, climate change, and food shortages, there is a need to find ways to grow more crops and develop new rice varieties that can cope with all the problems that come up. To contribute to this goal, we created a rice population using pollen from the iron-tolerant Suakoko 8 variety to improve the Bao Thai variety. For two years, we studied F5 and F6 populations that came from the above cross in Edozighi and Ibadan, which were different because the soils there were high in iron and low in iron, respectively. Following this phenotypic evaluation, different quantitative and qualitative data collected on the genotypes during these field experiments were analyzed for a deeper comprehension of the iron toxicity effect in the rice population developed on the one hand and the identification of iron-excess-tolerant genotypes in the lowlands [22,44], which impairs the expansion of rice cultivation in Africa [15,45]. This method contributed to the efficiency of the selection process and the rice breeding goals [46].

This study discovered that there was a significant variation between the genotypes studied for all the attributes evaluated. Statistical analyses revealed that those genotypes differed significantly depending on the phenology, with lines having early, medium, and late heading days. Plant materials with early and late maturity could be used for the desired maturity or based on the forecasted climate. Early maturity genotypes can avoid stress intervals that trigger drought avoidance and the harmful consequences of salt stress in rice [47,48]. The results revealed that Fe^2+^ toxicity delayed flowering in Edozighi compared to Ibadan. Correlation studies confirmed this pattern, revealing a slightly positive relationship between DH and LBS. The path analysis also corroborated this pattern, indicating a positive relationship between LBS and DH. Theerawitaya et al. [49] supported these findings by using genome-wide association analysis to identify features responding to iron toxicity stress at various stages in rice. According to these researchers, multiple QTLs in plants are linked to this mechanism.

The results demonstrated a high level of variation within the population for agronomic traits such as PH, NT, NP, PL, and GY. In Edozighi, the soil’s iron content significantly reduced PH, PL, and GY levels. Our findings are in line with the research investigations of Faruk et al. [50], who investigated various rice genotypes subjected to high iron levels in the soil. Our findings highlight a previously reported reduction in plant height caused by iron [51,52]. The negative effect of high iron on rice yields is consistent with earlier findings [15,17,50]. On the other hand, the current study found that the number of panicles and tillers per plant increased in the area with high iron levels in the soil. Correlation analysis, which demonstrated a favorable relationship between iron toxicity and an increase in the number of tillers and panicles per plant, was also supported by previous trends. Furthermore, path analysis revealed an identical attitude among the populations tested for iron toxicity tolerance. This group’s unusual behavior was comparable to an adaptation mechanism some species have developed to counteract the effects of stress. During this investigation in Edozighi, the increase in the number of tillers and panicles reported on the plants is circumstantial. As a result, the majority of these abnormally emitted tillers died before they reached maturity. During the harvest, too many panicles were stunted and sterile, which is similar to what Faruk et al. [50] found: 50% of the grain on each panicle was empty when they looked at the grain yield under Fe toxicity conditions. According to Arthaud [53], no organism can endure the whole range of ecological conditions on earth, and natural selection has resulted in a variety of adaptations to environmental limits. Adaptations are described as an organism’s traits that help it to survive or reproduce more effectively in its environment. The presence of these adaptations in an organism is limited by energetic trade-offs, as limited resources in the environment can only be allocated to particular types of adaptation. These modifications concern anatomical features, physiological processes, and behavioral mechanisms. Plants, being sessile organisms, are severely confined by environmental pressures and exhibit several adaptations when compared to animals, which, for the most part, have the ability to move throughout their environment to acquire the various necessary supplies.

The correlation results and path analysis showed that plant height and panicle length had a significant impact on yields. Farmers’ biggest concern is the grain yield because they need to sell it after harvesting. As a result, the breeder must prioritize these two characteristics when selecting the best line to reduce the negative impacts of iron toxicity on field yield. Our findings clearly highlight plant height and panicle length as essential components of rice productivity, and they corroborate various earlier studies. Li et al. [54] studied the relationships between yield and yield-related features for rice varieties released in China from 1978 to 2017. They found that plant height, number, and panicle length had a direct effect on yield, along with a number of other agronomic traits. Similarly, Bocco et al. [55] reported that plant height and panicle length per plant helped reduce rice losses during drought. On the other hand, Sikirou et al. [17] discovered that while working in pots on soils from diverse regions, plant height or number of panicles did not significantly affect grain yield. This difference between the findings, which stated that plant height and panicle length had a positive direct effect on yields, and the one that revealed that they had no influence on yields, may be due to the severity of the stress in situ. Similar results were found between grain yield and iron toxicity, while confirming that leaf bronzing score is a secondary trait of an indicator in selection for iron toxicity tolerance (18). Several genes, influenced by a wide range of variables, result in different behaviors, making all the above patterns typical. This study might have collected field data using only effective panicles and tillers. Thus, only tillers and panicles that have at least one well-filled seed would be considered. As a result, the tillers and panicles produced by the plant to avoid iron toxicity would not be considered during data collection. This research did not find a link between genes and the number of tillers and panicles per plant. This suggests that genotypes’ tendency to produce an excessive number of tillers and panicles may be due to high environmental variances. Furthermore, for this assessed population, taller, high-yielding plants under stress with good panicle length and well exerted would be preferable. A longer panicle has a great chance of possessing several branches and grains to increase yields.

The visualization of yields by genotype in the two areas revealed the amount of rice loss due to the high iron content of the soil. Grain yields in the two study contexts ranged from low to high (6–94% loss) due to the application of the Reyniers et al. [44] formula and some breeder decisions. Three lines, IR 88638-325-1-1-1-1-1, IR 88638-102-1-1-1-1-1, and IR 88638-34-1-1-1-1-1, have demonstrated consistent and stable yields throughout the trials, making them suitable for use as donors in the short term. High-performance lines can be developed or employed in a breeding program to combat this constraint in the medium and long term. Several lines showed very modest losses when exposed to iron toxicity, in contrast to the tolerant parent, Suakoko 8. Similarly, some offspring generated more than both parents combined. The population behavior reflects the heterosis effect evident in multiple generations, resulting in enhancements in agronomic traits. Our findings are supported by previous research on controlling rice constraints, particularly iron toxicity, weeds, drought, blasts, submergence, and gall midges [17,55,56,57,58,59]. These researchers found resistant varieties and crossed them with susceptible parents to create more resistant lines. Nowadays, all researchers agree that genetic improvement of plants is one of the most effective ways to develop plants that are resistant to various constraints.

## 5. Conclusions

This assessed population exhibited significant variation, with genotypes ranging in plant height, number of tillers, number of days at 50% heading, panicle length, and grain production (kg/ha). This difference, as revealed by variance analysis, is inherited and derives from both parents’ sides as a result of gene exchange. The findings demonstrated a negative association between iron toxicity and plant height, panicle length, and grain yield. Conversely, this study found a link between the leaf bronzing score and the number of days at 50% plant heading, the number of panicles, and the number of tillers per plant. Grain yield was positively correlated with plant height and panicle length. Grain yield was negatively correlated with the quantity of tillers and panicles per plant. This final remark is a translation of the plant’s survival mechanism, which involves initiating complementary organs to resist stress, because these characteristics are stated as yield components. Path analysis demonstrated that a high iron toxicity had a direct and positive effect on the number of days at the 50% heading period, a direct and negative effect on plant height, a direct and positive effect on the number of tillers and panicles, and a direct but negative effect on panicle length. The clustering analysis identified four groups with distinct traits. Excess iron in the soil is expected to cause losses in yield ranging from 6 to 94%. The authors suggest that the three resilient lines (IR 88638-325-1-1-1-1-1, IR 88638-102-1-1-1-1, and IR 88638-34-1-1-1-1) be used as donors because they lose less yield by 6, 7, and 7, respectively. Our findings suggest immediate, short-term, and long-term strategies for research and mitigating iron toxicity in rice farming. Additional research would be required to corroborate the performance described in this study.

## Figures and Tables

**Figure 1 plants-13-01610-f001:**
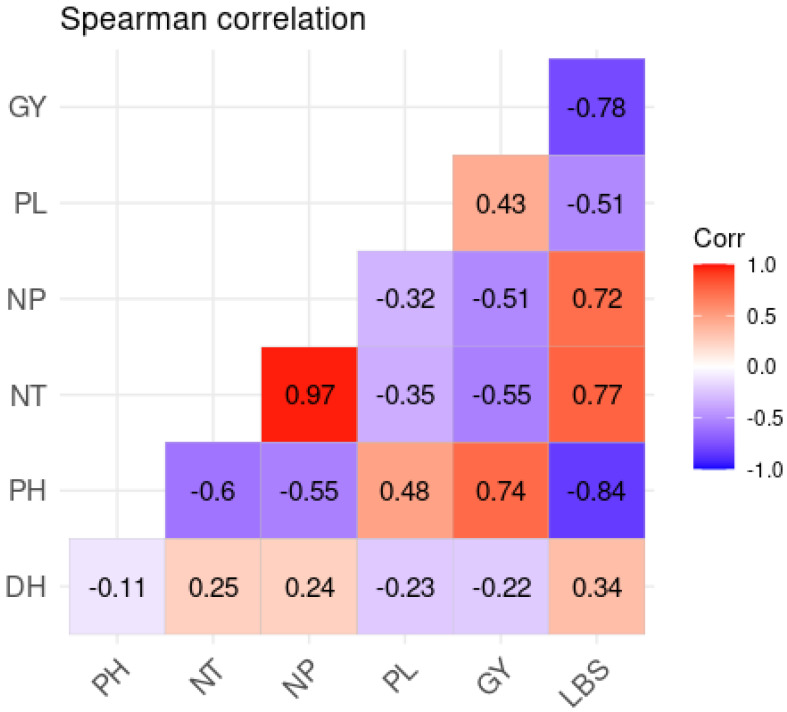
Correlations between the number of days to heading (DH), plant height (HP), number of tillers (NT), panicle number (NP), panicle length (LP), yield (GY), and leaf bronzing score (LBS).

**Figure 2 plants-13-01610-f002:**
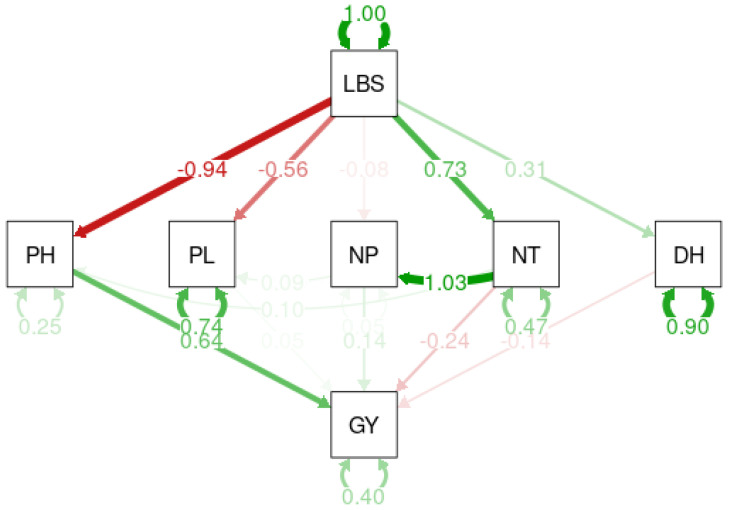
Path diagram between the number of days to heading (DH), plant height (HP), number of tillers (NT), panicle number (NP), panicle length (LP), grain yields (GY), and the leaf bronzing score (LBS).

**Figure 3 plants-13-01610-f003:**
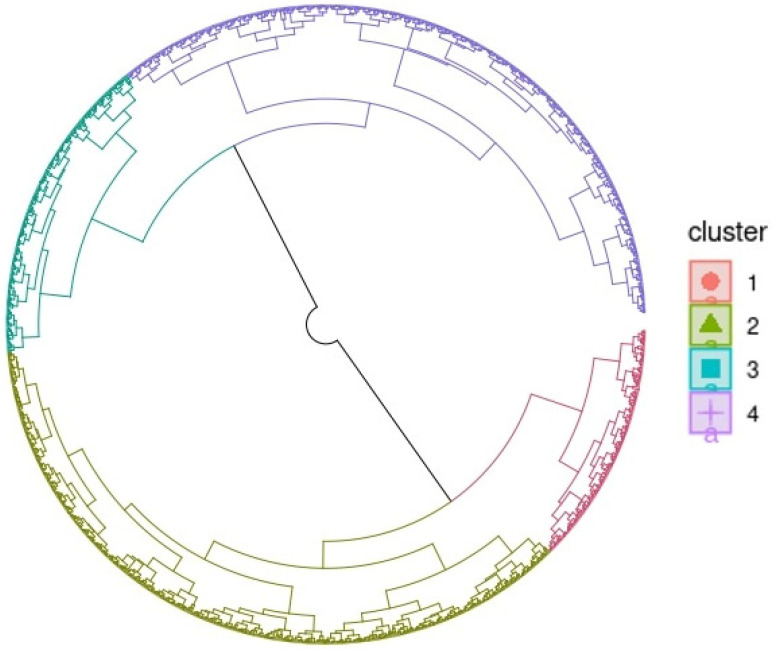
Principal component analysis and characterization of genotype groups.

**Figure 4 plants-13-01610-f004:**
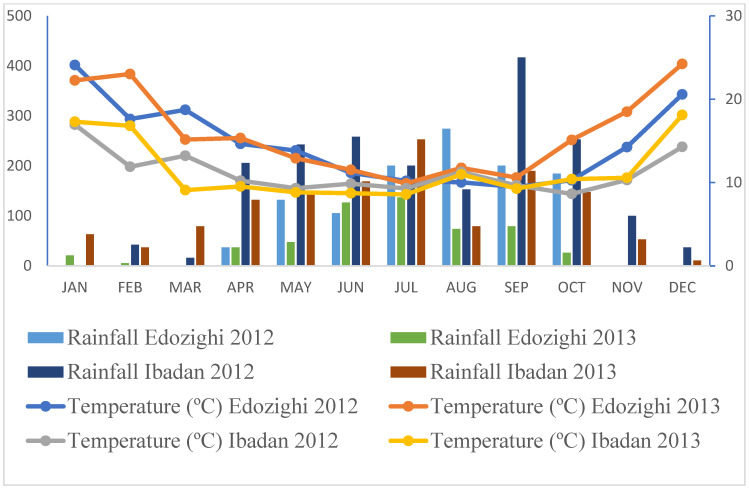
Monthly rainfall (mm) and temperature (°C) recorded at the experimental sites of Edozighi and Ibadan in 2012 and 2013 (source: IITA Ibadan).

**Figure 5 plants-13-01610-f005:**
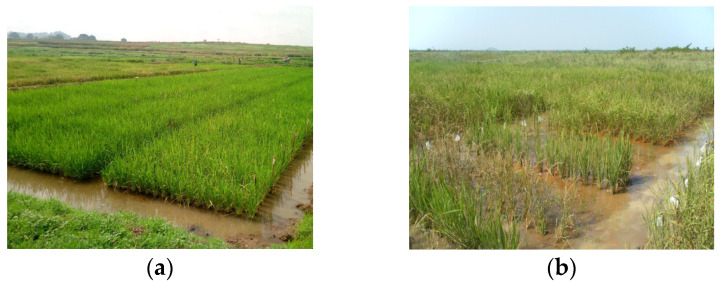
(**a**) Field at Ibadan. (**b**) Field at Edozighi.

**Table 1 plants-13-01610-t001:** Significance test based on Kruskal–Wallis test medians for number of days to heading, plant height (PH), number of tillers per plant (NT), number of panicles per plant (NP), panicle length (PL), and yield by genotype (GY).

Traits	Mean Sq	Df	Pr (>F)
Days to heading	153	1	0.001 ***
Plant height	967	1	0.001 ***
Number of tillers per plant	811	1	0.001 ***
Number of panicles per plant	702	1	0.001 ***
Panicle length	350	1	0.001 ***
Grain yield	835	1	0.001 ***

***: Highly significant.

**Table 2 plants-13-01610-t002:** Variations in growth characteristics according to study environments.

Traits	Control Treatment (Ibadan) (N = 680)	Site with High Iron Content (Edozighi) (N = 680)	Overall (N = 1360)
Days to heading			
Mean (SD)	85.7 (8.79)	91.8 (9.85)	88.7 (9.82)
Median [Min; Max]	87.0 [62.0; 110]	94.0 [69.0; 116]	90.0 [62.0; 116]
Plant height			
Mean (SD)	160 (17.5)	108 (12.6)	134 (30.5)
Median [Min; Max]	163 [102; 201]	108 [58.3; 148]	127 [58.3; 201]
Number of tillers			
Mean (SD)	11.0 (2.97)	19.5 (4.82)	15.2 (5.85)
Median [Min; Max]	10.7 [5.00; 23.0]	18.8 [10.0; 36.3]	14.3 [5.00; 36.3]
Number of panicles			
Mean (SD)	9.47 (2.80)	16.5 (4.69)	13.0 (5.21)
Median [Min; Max]	9.00 [3.33; 26.7]	15.7 [6.67; 34.3]	12.0 [3.33; 34.3]
Panicle length			
Mean (SD)	28.8 (3.17)	25.2 (2.99)	27.0 (3.57)
Median [Min; Max]	29.0 [20.0; 41.0]	25.0 [18.3; 32.0]	27.0 [18.3; 41.0]
Grain yield			
Mean (SD)	6120 (2520)	1390 (665)	3750 (3000)
Median [Min; Max]	6230 [283; 11,700]	1340 [0; 8230]	2240 [0; 11,700]

**Table 3 plants-13-01610-t003:** Ranking of the top ten genotypes with the lowest grain yield loss in Edozighi and Ibadan.

Genotypes	Edozighi GY	Ibadan GY	%Losses
IR 88638-325-1-1-1-1-1-1	956	1015	6
IR 88638-102-1-1-1-1-1-1	640	685	7
IR 88638-34-1-1-1-1-1-1	1608	1732	7
IR 88638-98-1-1-1-1-1-1	1904	2718	30
IR 88638-240-1-1-1-1-1-1	1187	1765	33
IR 88638-13-1-1-1-1-1-1	1430	2352	39
IR 88638-230-1-1-1-1-1-1	2855	4793	40
IR 88638-308-1-1-1-1-1-1	1537	2765	44
IR 88638-39-1-1-1-1-1-1	2803	5223	46
IR 88638-208-1-1-1-1-1-1	1299	2431	47
Suakoko 8 (parent line)	987	2467	60
BW348-1	1295	4035	68
WITA4	1277	4420	71
Bao Thai (parent line)	915	4083	78
IR64	620	3439	82

## Data Availability

Data are provided with this article and Appendix A.

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
