# Peer review of "Field Evaluation of Rice Lines Derived from Suakoko 8 X Bao Thai for Iron Tolerance in the South Saharan African Farming System"

_plants, 2024, doi:10.3390/plants13121610_

Round 1
Reviewer 1 Report
Comments and Suggestions for Authors
In the manuscript “Field Evaluation of Rice Lines Derived from Suakoko 8 X Bao Thai for Iron Tolerance in the South Saharan African Farming System”, the authors investigated the phenotypes and yield formation of 340 rice introgression lines derived from a cross between Suakoko and Bao Thai in Edozighi and Ibadan, Nigeria, which are different from iron levels in the soil. The manuscript is meaningful, and could generate some varieties that can be potentially cultivated in iron-rich soils. However, there are some shortcomings that need to be solved.
1. The authors indicated the characteristics of rice varieties Suakoko 8 (resistant) and Bao Thai (susceptible), then where can we get the convincing description. Are there any data, references, or experimental data and phenotypes to favor the statement?
2. In addition to the basic data from field cultivation, did the authors also carry out some experiments, like iron level in the two regions, Fe accumulation in different lines, different tissues, and different regions, the potential molecular mechanism…
3. The results from the current work can be concluded easily, seeming like a basic description for an early work in breeding. Therefore, the authors should supplement more experiments to get a more meaningful concept for the work.
4. The result part seems like a repetition of the result, more deep discussion should be provided to take the findings of the work further.
5. The two regions Edozighi and Ibadan, Nigeria may have different climate conditions during the two years. How did the authors regulate the variations except various lines and different soils.
Minor:
1. Abstract, line 24: “Two-year experiments employed an alpha-lattice design were completed”, the sentence should be rewritten.
Comments on the Quality of English LanguageSome minor errors exist in the current form.
Author Response
Dear reviewer,
We appreciate your comments on the original manuscript. We herewith would like to submit the revised draft to you with the list of our responses (see attached – in the table) to the comments made by you. Similarly, we highlighted the text where we changed in tracking in the revised manuscript. We feel that the revised version of the paper is much clearer than the original version.
Finally, we would like to express our sincere appreciation to you for carefully checking the draft and providing us with constructive comments and suggestions.
Thank you,

Reviewer 2 Report
Comments and Suggestions for Authors
The current study explored high-yielding and Fe-toxicity-tolerant irrigated lowland 94 rice among a population derived from crossing Suakoko 8 and Bao Thai under field con- 95. Their results revealed that multiple characteristics had both direct and indirect effects on cultivated rice yields. The findings from this study showed significant changes in yield and yield characteristics between genotypes. Grain yields ranged from 283 to 11700 kg/ha in the absence of iron in the soil, contrary to 0 to 8230 kg/ha in soil with iron toxicity, with losses estimated between 6 and 94%, demonstrating the resulting disaster. In contrast to the elite parents and varieties used in this study, the ten top genotypes exhibited smaller losses in yield. The authors strongly recommend using these lines for further studies as donors or releasing them in farmer fields in Africa.
In general, the paper was well-written, the paper had the correct experimental design and produce significant results, I have the following comments:
Key comments:
1. The data of presentation is difficult to understand, it does not show the actual data from different rice lines, only one Table to show the top ten genotypes with the lowest grain yield loss
Minor Comments:
1. Abstract should mention the objectives of the study. The current study investigated 340 rice introgression lines derived from a cross between Suakoko 8 and Bao Thai in Edozighi and Ibadan what purpose?
2. It should have at least one figures to show how the plants look like, it only has text and data?
3. Table 1 and Table 2 should show the data for each genotypes?
4. There is no reference cited from 2024. Literature need to be updated.
Author Response
Dear reviewer,
We appreciate your comments on the original manuscript. We herewith would like to submit the revised draft to you with the list of our responses (see attached – in the table) to the comments made by you. Similarly, we highlighted the text where we changed in tracking in the revised manuscript. We feel that the revised version of the paper is much clearer than the original version.
Finally, we would like to express our sincere appreciation to you for carefully checking the draft and providing us with constructive comments and suggestions.
We hope that you will appreciate its current writing and the point-by-point answers provided below.
Regards,

Round 2
Reviewer 1 Report
Comments and Suggestions for Authors
The manuscript has been improved a lot, and my concerns have been responded properly.
Reviewer 2 Report
Comments and Suggestions for Authors
Since the authors have addressed all my questions, thus, I have no further comments.